# Short-Term and Long-Term Outcomes of Fetal Ventriculomegaly beyond Gestational 37 Weeks: A Retrospective Cohort Study

**DOI:** 10.3390/jcm12031065

**Published:** 2023-01-30

**Authors:** Huiling Chen, Peng Bai, Shuqi Yang, Mingzhu Jia, Huan Tian, Juan Zou, Xue Xiao

**Affiliations:** 1Key Laboratory of Birth Defects and Related Diseases of Women and Children (Sichuan University), Ministry of Education, Chengdu 610041, China; 2Department of Gynecology and Obstetrics, West China Second University Hospital, Sichuan University, Chengdu 610041, China; 3Department of Forensic Genetics, West China School of Basic Medical Sciences & Forensic Medicine, Sichuan University, Chengdu 610041, China; 4Department of Pathology, West China Second University Hospital, Sichuan University, Chengdu 610041, China

**Keywords:** ventriculomegaly, short-term, long-term, NICU, width, reproductive outcome

## Abstract

Birth defects have brought about major public health problems, and studying the clinical outcomes of the most common prenatal central nervous system abnormality, namely, fetal ventriculomegaly (VM), is helpful for improving reproductive health and fertility quality. This is a retrospective cohort study from 2011 to 2020 in the West China Second University Hospital, Sichuan University, aiming to evaluate the short-term and long-term outcomes of VM over 37 weeks’ gestation to exclude the influence of preterm birth. The study analyzed data from 401 term pregnancies, with 179 VM and 222 controls. From the short-term outcomes, the rate of the neonatal intensive care unit (NICU) admission under the VM group (10.06%) was comparatively higher than the control (0.45%), but Apgar scores between both groups at 1 min, 5 min and 10 min were not significantly different. From the long-term outcomes, there were more infants with abnormal neurodevelopment under the VM group than control (14.53% vs. 2.25%, *p* < 0.001). In addition, NICU admission (*p* = 0.006), peak width of lateral ventricles (*p* = 0.030) and postnatal cranial ultrasound suggestive with VM (*p* = 0.002) were related to infants’ long-term outcomes. NICU admission during the perinatal period was an independent risk factor for the adverse long-term outcomes (OR = 3.561, 95% CI 1.029–12.320, *p* = 0.045). In conclusion, VM impairs short-term and long-term outcomes of term infants. Short-term outcome, especially NICU admission, could predict their adverse long-term outcomes.

## 1. Introduction

Birth defects have brought a great burden to individuals and society, and fetal ventriculomegaly (VM) is the most common central nervous system abnormality identified by prenatal ultrasound [1,2]. VM is defined as the width of the lateral ventricle ≥10 mm, according to the reference range of ventricle width. [3,4]. Many causes account for VM, including normal physiological variations, abnormality of cerebrospinal fluid reflux due to a variety of pathological factors [2,5,6], intracranial manifestation of some systemic diseases, chromosomal abnormalities and viral infections [7], among others. VM may cause a variety of adverse outcomes, especially neurological, motor and /or cognitive defects [8,9]. Careful consideration of the outcomes of VM can comprehensively improve the level of reproductive health and improve the quality of the birth population.

There have been a variety of studies about the prognosis of VM, many of which often evaluated the outcomes of all infants born at different gestational weeks [7], ignoring the effect of preterm delivery, which would be a great confounding factor [10]. It is challenging to synthesize the reported data due to different gestational ages. Typically, gestation age had profound effects on the infants’ outcomes. Accordingly, this retrospective cohort study was particularly focused on term pregnancies to evaluate the short-term and long-term outcomes of VM to avoid the effect of preterm birth, which is expected to provide guidance for clinical practices involved in VM.

## 2. Designs and Methods

A retrospective cohort study was performed at the Second West China Hospital of Sichuan University from March 2011 to September 2020. Term pregnancies with VM during the period formed the VM group. The control group, term pregnancies without VM, were randomly selected for each year with a comparable number of both VM and control groups maintained for each year to ensure consistent medical conditions and follow-up times for all subjects in the study. Women who did not deliver at our hospital were excluded, and infants without follow-up after birth were excluded from the analysis. This study was approved by the ethics committee of West China Second Hospital of Sichuan University. 

Maternal demographic data, pregnancy, delivery and neonatal information were extracted from the electronic medical records. VM was defined as the width of the lateral ventricle ≥ 10 mm detected by ultrasonic scan and/or fetal magnetic resonance imaging (MRI). For the VM group, additional details on VM were recorded (i.e., whether bilateral or unilateral VM, first scanning ventricular width, gestational age of first scanning, peak ventricular width and last scanning ventricular width). 

VM were categorized into mild (10–12 mm), moderate (12.1–15 mm) and severe (> 15 mm) degrees based on the peak ventricular width. According to whether other structural anomalies or any chromosomal abnormalities existed, VM were also divided into isolated VM and non-isolated VM. Comparing the width of the lateral ventricle at the time of initial diagnosis with that of the last ultrasound examination aiming at the lateral ventricle, VM were further divided into progressing (width of the lateral ventricle is increasing ≥ 2.0 mm compared with the initial diagnosis) and not-progressing (width of the lateral ventricle is decreasing or increasing < 2.0 mm) groups. 

The time horizon of the short-term and long-term outcomes for VM was not clearly defined to the best of our knowledge. Most current studies about outcomes of fetal neurodevelopment focused on the perinatal period and 1 month to 2 years postpartum [11,12,13,14]. In effect, the short-term outcomes were defined as Apgar scores and the NICU admission during the perinatal period in this study. All subjects were followed up regarding the development of the fetuses’ nervous system, including conditions of intelligence, motor and language beyond 1 year after birth. Long-term outcomes, as opposed to short-term outcomes, were assessed as any clinical manifestations of abnormal neurodevelopment occurring after 1 year of birth in this study.

Statistical analysis was performed using SPSS software (IBM SPSS Statistics 26, Chicago, IL, USA). Quantitative data were described using means and standard deviations, while categorical data were reported by the number of events. Mean comparison among independent groups was evaluated using Student’s *t* test. Categorical data were comparatively assessed using Pearson’s chi-squared test, Fisher’s exact test or Mann–Whitney test. A two-tailed *p* value < 0.05 was considered statistically significant. The sample size of 160 participants per group was evaluated to have 90% power with a two-sided alpha of 0.05, based on an incidence of adverse long-term outcomes of 12% in VM with about 2% in the baseline population or an incidence of 10% in VM with 1% in the baseline population. To accommodate 15% attrition, a goal of 200 pregnancies in each group was targeted.

## 3. Results

The study included 463 term pregnancies from March 2011 through September 2020, with VM in 201 pregnancies diagnosed by ultrasound and/or fetal MRI (Figure 1), and 262 pregnancies in the control group. Of these, 22 (10.95%) subjects in the VM group and 40 (15.4%) subjects in the control group were lost to follow-up (*p* = 0.176). Finally, a total of 401 participants, 179 with VM and 222 without VM, were included in the study (Figure 2). 

The general conditions of the two groups did not show any statistically significant differences, as summarized in Table 1. Pregnancy with complications, namely, pregnancy with diabetes mellitus, premature rupture of membrane, pregnancy with thyroid disorder, and pregnancy associated with cardiac disease, were counted together and were comparable between the VM and control groups.

For the short-term outcome of infants as shown in Table 2, the rate of the NICU admission in the VM group (10.06%) was comparatively higher than the rate from the control group (0.45%), although the Apgar scores between both groups at 1 min, 5 min and 10 min were not statistically significant. In addition, there seems to be more male and larger-sized infants in the VM group than in the control group.

Notably, the difference in the long-term outcome between both groups was statistically significant during the follow-up. There were more infants with abnormal neurodevelopment in the VM group than the control group (14.53% vs. 2.25%, *p* < 0.001). In addition, a stratified analysis for the long-term outcome based on gestational age at birth (37 0/7–38 6/7 weeks vs. ≥39 0/7 weeks) was also performed. It showed no significant difference in the gestational age between the two groups (*p* > 0.05).

Comparing the characteristics between the VM subgroups as summarized in Table 3, the adverse long-term outcome rates of mild, moderate and severe VM were 10.95% (15/137), 22.50% (9/40) and 100% (2/2), respectively, and these differences were determined to be statistically significant (*p* < 0.05). Furthermore, the pairwise comparison of prognosis among mild, moderate and severe VM was performed by Bonferroni post hoc analysis, and *p* < 0.0167(0.05/3) was considered statistically significant. Pairwise comparisons showed no statistical differences between mild VM and moderate VM (*p* > 0.0167), as well as between moderate and severe VM (*p* > 0.0167). However, the difference in prognosis rates between mild and severe VM was found to be statistically significant (*p* < 0.0167), while infants’ short-term outcomes, referring to the NICU admission, was associated with adverse long-term outcomes (*p* = 0.006). Furthermore, if postnatal cranial ultrasound showed manifestations of VM, the prognosis was more likely to be poor (*p* = 0.002). However, other characteristics about VM, such as isolated/non-isolated VM (*p* = 0.116), unilateral/bilateral VM (*p* = 0.260), and variation of width (*p* = 0.240), did not seem to be relevant to the long-term outcome. 

A binary logistic regression analysis was performed to evaluate the significance of the various clinicopathological factors considered in the study in predicting the long-term outcomes of VM. Results presented in Table 4 showed that NICU admission during the perinatal period was an independent risk factor for the adverse long-term outcomes (OR = 3.561, 95% CI 1.029–12.320, *p* = 0.045). Other factors, such as the peak width of lateral ventricles, and gestational age of delivery, were not independent risk factors for the long-term outcomes, as noted from the analysis.

## 4. Discussion

Birth defects have brought about major public health problems, and to study the clinical outcomes of VM is helpful for clinical decisions, thereby improving reproductive health and fertility quality. In this retrospective cohort study, we discovered that short-term and long-term outcomes of infants with VM were affected negatively to some extent. In the short term, the rate of NICU admissions in the VM group (10.06%) was comparatively higher than the control (0.45%). However, the Apgar between both groups were not significantly different. The reason for the higher NICU admission rate in VM may be due to the predisposition of children with VM to show some signs of increased intracranial pressure after birth, including reduced activity, vomiting, widening of the fontanelle and apnea, among others [15]. In practical clinical work, neonates with VM during the perinatal period would be transferred to NICU because of the symptoms mentioned above for further monitoring. As observed, a small number of newborns were in fact transferred to NICU because of VM alone, rather than other manifestations. All these may account for the higher rate of the NICU admission.

Notably, the difference in the long-term outcome between both groups was statistically significant during the follow-up. VM infants were more likely to manifest abnormal neurodevelopment after birth as compared with the control group. Existing studies have reported that VM can lead to neurodevelopmental disorders of children after birth in movement, language, cognition and other aspects, and may also cause some neuropsychiatric diseases such as autism, schizophrenia, epilepsy, attention deficit/hyperactivity disorder and so on [16,17]. As is known, gestational age greatly influences fetal outcomes [10]. Therefore, to avoid the effects of gestational age, our study focused on evaluating outcomes of pregnancies beyond 37 weeks. In addition, a stratified analysis based on gestational age at birth (37 0/7–38 6/7 weeks vs. ≥39 0/7 weeks) was also performed, and the result showed similar long-term outcomes in the different gestational ages.

Interestingly, VM seems to have occurred in more male and larger-sized infants than in the control group. In 1994, Patel et al. noted that there were more male than female fetuses with isolated mild VM (*p* < 0.05) [18]. In addition, studies have suggested that the lateral ventricles of male fetuses were relatively larger than those of female fetuses (6.7 ± 1.3 mm vs. 6.3 ± 1.4 mm, *p* < 0.001) [19]. Therefore, it was hypothesized that the basal value of the lateral ventricles of male fetuses is higher than those of female fetuses, so under the effect of pathological factors, male fetuses may be more likely to show VM than female fetuses. However, there was no significant correlation between gender and outcomes, and our research is consistent with previous studies [20]. Based on clinical experience, male fetuses tend to be heavier and larger, which could account for larger-sized infants in the VM group than the control group. 

Our study further analyzed the factors associated with the long-term prognosis of VM, which suggested NICU admission and degrees of VM were related to infants’ long-term outcomes. Infants transferred to the NICU may have some manifestations of brain injuries during early birth, which may have implications for their long-term neurodevelopment. Typically, VM’s outcome depends on its cause and degree [21]. In 2018, SMFM (Society for Maternal-Fetal Medicine) pointed out that the likelihood of normal neurodevelopment for mild VM, especially the isolated VM, is more than 90%, while moderate VM is 75–93% [22]. Studies have also shown that severe VM, indicated by the width of lateral ventricle being ≥15 mm, often predicts worse fetal prognosis [23,24]. In this study, the proportion of normal neurodevelopment among mild, moderate and severe VM was 89.05% (122/137), 77.50% (31/40) and 0% (0/2), respectively, which is largely consistent with previous studies [22,25,26]. Further pairwise comparison showed that the difference between mild and severe VM was significant (*p* < 0.0167); although notably, logistic regression analysis showed that peak width of lateral ventricle was not an independent risk factor for adverse long-term outcomes (OR = 1.429, 95% CI 0.935–2.185, *p* = 0.099). However, only two cases of severe VM were included in this study. The reason may be that pregnancies with severe VM generally chose to terminate pregnancy. Therefore, prospective studies with a larger sample size of severe VM are needed to evaluate the prognosis.

The study has some limitations. First, due to limited follow-up time for some infants, some clinical symptoms of the infants may not have been fully traced, with the actual incidence of poor prognosis potentially under-reported. Besides, the poor prognosis of VM sometimes manifests within one year after birth but some may not manifest until school age [14]. The short-term outcomes defined roughly as the NICU admission and Apgar may overstate the conclusion. Second, due to the study design being essentially of cohort type, it might have been subject to selection bias and unmeasured confounders, although it was reassuring that nearly all patient and pregnancy characteristics were similar between groups. During the follow-up of the delivered fetus, the subjective description from the family and the pediatrician’s evaluation results were used for the long-term outcomes of the fetus. The family may have been hesitant to report the real facts of the fetus because of the poor prognosis, or the parents may have underestimated the abnormal performance of the fetus, resulting in potentially biased outcomes.

In conclusion, VM may have negatively affected the short-term and long-term outcomes of infants. These findings highlight the need for additional care for VM infants, especially with the NICU admission after birth.

## Figures and Tables

**Figure 1 jcm-12-01065-f001:**
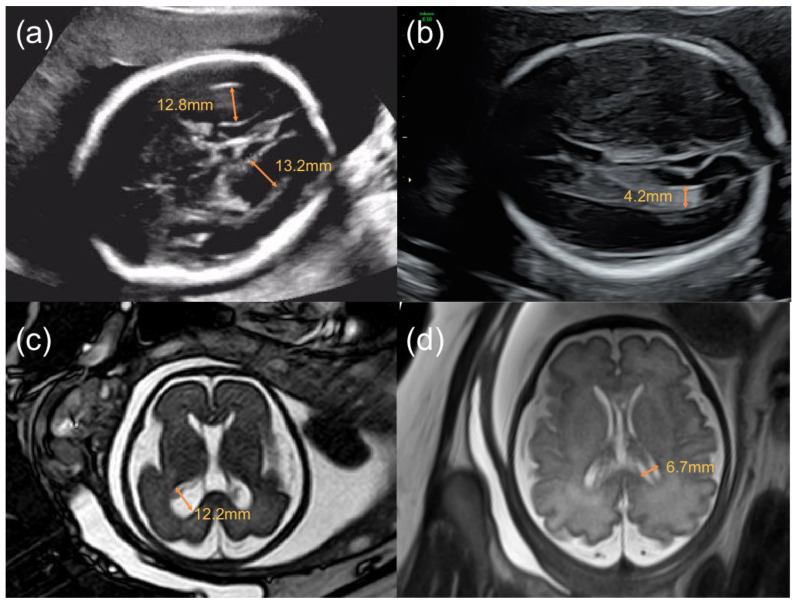
Images of transabdominal ultrasound (**a**,**b**) and fetal MRI (**c**,**d**) demonstrating normal and abnormal lateral ventricles. (**a**). Axial ultrasound image at 27 4/7 gestational weeks, singleton pregnancy, showing fetal bilateral ventriculomegaly with left lateral ventricle = 12.8 mm, right lateral ventricle = 13.2 mm; (**b**). Axial ultrasound image at 23 6/7 gestational weeks, singleton pregnancy, showing normal lateral ventricle; (**c**). Axial T2 fetal MR image at 27 3/7 gestational weeks, singleton pregnancy, showing fetal unilateral ventriculomegaly with left lateral ventricle = 12.2 mm; (**d**). Axial T2-SSFSE (single-shot fast spin echo) fetal MR image at 32 gestational weeks, singleton pregnancy, showing normal lateral ventricle.

**Figure 2 jcm-12-01065-f002:**
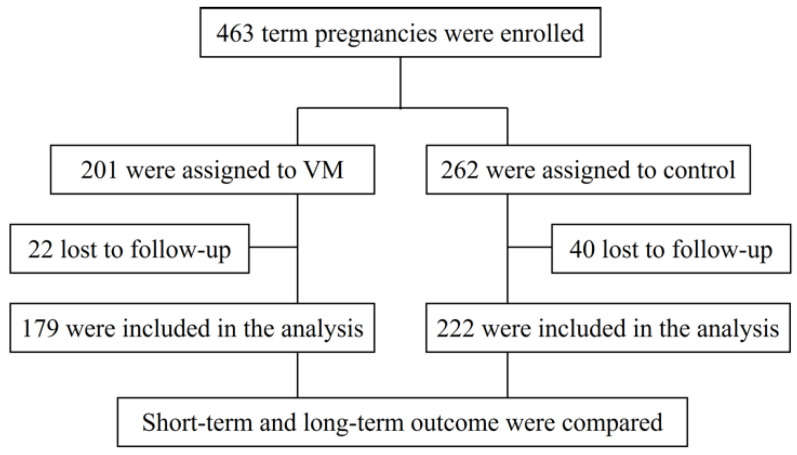
Profile of the cohort study. The short-term outcomes of infants were measured primarily using Apgar scores and the NICU admission during the perinatal period while long-term outcome was assessed as any clinical manifestations of abnormal neurodevelopment occurring after 1 year of birth.

**Table 1 jcm-12-01065-t001:** General conditions of VM group and control group.

Variable	VM Group (*n* = 179)	Control Group (*n* = 222)	*p* Value
Maternal age/(year)	31.06 ± 4.05	31.30 ± 3.78	0.531
Pre-BMI/(kg/m^2^) ^a^	20.63 ± 2.63	20.84 ± 2.69	0.447
Gravidity	2.34 ± 1.38	2.19 ± 1.43	0.302
Parity	1.36 ± 0.49	1.39 ± 0.53	0.564
Pregnancy with complications (n./total n.) (%)	97/179(54.19)	103/222(46.40)	0.121
IVF-ET(n./total n.) (%)	15/179(8.38)	11/222(4.95)	0.166
Twin pregnancy(n./total n.) (%)	7/179(3.91)	14/222(6.31)	0.284
TORCH IgM positive of maternal serum (n./total n.) (%) ^b^	7/49(14.29)	4/29(13.79)	1.000 *
Abnormal chromosomal analysis of amniocentesis (n./total n.) (%) ^c^	8/92(8.70)	2/45(4.44)	0.497 *
Cesarean Section (n./total n.) (%)	116/179(64.80)	140/222(63.06)	0.718

^a^ 1 VM without pre-BMI record; ^b^ 49 VM and 29 control were given a blood TORCH antibody test; ^c^ 92 VM and 45 control underwent amniocentesis for fetal karyotyping or chromosome microarray analysis; *p* values were calculated with Student’s *t* test or Pearson’s chi-squared test(two-sided), except as * *p* values were calculated with Fisher’s exact test (two-sided).

**Table 2 jcm-12-01065-t002:** Short-term and long-term outcomes of the VM group and control group.

Category	VM Group (*n* = 179)	Control Group (*n* = 222)	*p* Value
Gestational age at birth/(weeks)	38.72 ± 1.05	38.73 ± 1.08	0.925
Distribution(n./total n.) (%)			0.902
37 0/7–38 6/7 weeks	68/179(37.99)	83/222(37.39)	
≥39 0/7 weeks	111/179(62.01)	139/222(62.61)	
Male infant (n./total n.) (%)	110/179(61.45)	104/222(46.85)	0.004
Length at birth/mm ^a^	49.89 ± 2.01	49.32 ± 2.71	0.020
Birth weight/g	3382.99 ± 477.28	3233.18 ± 425.10	0.001
Apgar at 1 min	9.90 ± 0.38	9.95 ± 0.35	0.168
Apgar at 5 min	9.99 ± 0.08	9.99 ± 0.12	0.428
Apgar at 10 min	10.00 ± 0.00	10.00 ± 0.00	1.000
NICU admission (n./total n.) (%)	18/179(10.06)	1/222 (0.45)	<0.001
Abnormal neurodevelopment after birth (n./total n.) (%)	26/179(14.53)	5/222(2.25)	<0.001

^a^: 1 VM and 1 control without infant length at birth records; *p* values were calculated with Student’s *t* test or Pearson chi-squared test (two-sided).

**Table 3 jcm-12-01065-t003:** Factors associated with long-term prognosis of live birth VM fetuses.

Factor	VM with Normal Neurodevelopment (*n* = 153)	VM with Abnormal Neurodevelopment (*n* = 26)	*p* Value
Maternal age/(year)	31.07 ± 4.044	30.96 ± 4.142	0.898
Pre-BMI/(kg/m^2^)	20.78 ± 2.65	19.79 ± 2.36	0.078
Gravidity	2.30 ± 1.396	2.54 ± 1.272	0.417
Parity	1.35 ± 0.491	1.42 ± 0.504	0.464
Pregnancy with complications (n./total n.) (%)	71/153(46.41)	12/26(46.15)	0.981
Twin pregnancy (n./total n.) (%)	6/153(3.92)	1/26(3.85)	1.000
IVF-ET (n./total n.) (%)	15/153(9.80)	0/26(0.00)	0.132 *
Male infant (n./total n.) (%)	98/153(64.05)	12/26(46.15)	0.083
Abnormal chromosomal analysis of amniocentesis ^a^	7/79(8.86)	1/13(7.69)	1.000 *
Gestational age at birth (weeks)	38.69 ± 1.053	38.85 ± 1.047	0.493
Distribution (n./total n.) (%)			0.209
37 0/7–38 6/7 weeks	61/153(39.97)	7/26(26.92)	
≥ 39 0/7 weeks	92/153(60.13)	19/26(73.08)	
Cesarean section (n./total n.) (%)	96/153(62.75)	20/26(76.92)	0.162
NICU admission (n./total n.) (%)	11/153(7.19)	7/26(26.92)	0.006 *
Non-isolated VM (n./total n.) (%)	52/153(33.99)	13/26(50.00)	0.116
Do MRI screen (n./total n.) (%)	94/153(61.44)	17/26(65.38)	0.701
Postnatal cranial ultrasound suggestive with VM ^b^	18/79(22.78)	8/11(72.73)	0.002 *
Bilateral VM (n./total n.) (%)	37/153(24.18)	9/26(34.62)	0.260
Gestational age at first diagnosis			
≥ 28 week (n./total n.) (%)	79/153(51.63)	14/26(53.85)	0.853
Progressing VM (n./total n.) (%)	11/153(7.19)	4/26(15.38)	0.240 *
Peak width of lateral ventricles (mm)	11.39 ± 1.046	12.25 ± 1.87	0.030
Degrees (n./total n.) (%)			0.029 **
Mild	122/153(79.74)	15/26(57.69)	0.070 ^c^
Moderate	31/153(20.26)	9/26(34.62)	0.064 ^d^*
Severe	0/153(0.00)	2/26(7.69)	0.014 ^e^*

^a^: amniocentesis for fetal karyotyping or chromosome microarray analysis was available in 92 of the cases; ^b^: postnatal cranial ultrasonography was performed in 81 cases with normal neurodevelopment and 11 cases with abnormal neurodevelopment; ^c^: compared with moderate VM, *p* > 0.0167; ^d^: compared with severe VM, *p* > 0.0167; ^e^: compared with mild VM, *p* < 0.0167; *p* values were calculated with Student’s *t* test or Pearson chi-squared test (two-sided), except as * *p* values were calculated with Fisher’s exact test (two-sided) and ** *p* value were calculated with Mann–Whitney test.

**Table 4 jcm-12-01065-t004:** Binary logistic regression analysis of long-term prognosis of live birth VM fetuses.

Factor	Category	B	S.E.	Wald	df	P	ExpB	95%CI
Pregnancy with complications	Yes	−0.268	0.494	0.294	1	0.587	0.765	0.290–2.015
Twin pregnancy	Yes	0.063	1.281	0.002	1	0.961	1.065	0.087–13.102
Male infant	Yes	−0.739	0.486	2.316	1	0.128	0.477	0.184–1.237
Gestational age at first diagnosis (weeks)	/	−0.065	0.056	1.345	1	0.246	0.937	0.839–1.046
Progressing VM	Yes	−0.157	0.829	0.036	1	0.850	0.855	0.168–4.342
Non-isolated VM	Yes	0.518	0.502	1.067	1	0.302	1.679	0.628–4.487
Bilateral VM	Yes	−0.137	0.567	0.059	1	0.809	0.872	0.287–2.651
Peak width of lateral ventricles (mm)	/	0.357	0.217	2.714	1	0.099	1.429	0.935–2.185
Gestational age at birth (weeks)	/	0.216	0.231	0.875	1	0.350	1.242	0.789–1.954
Cesarean section	Yes	0.728	0.565	1.660	1	0.198	2.070	0.684–6.264
NICU admission	Yes	1.270	0.633	4.023	1	0.045	3.561	1.029–12.320
Postnatal cranial ultrasound	No VM detected	/	/	3.812	2	0.149	/	/
	VM detected	1.590	0.818	3.780	1	0.052	4.903	0.987–24.350
	Not carried out	1.049	0.706	2.211	1	0.137	2.856	0.716–11.390
Constant	/	−13.766	9.454	2.120	1	0.145	0.000	/

## Data Availability

The data presented in this study are available on request from the corresponding author. The data are not publicly available, because the raw data required to reproduce these findings cannot be shared at this time as the data also forms part of an ongoing study.

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
