# Peer review of "Short-Term and Long-Term Outcomes of Fetal Ventriculomegaly beyond Gestational 37 Weeks: A Retrospective Cohort Study"

_jcm, 2023, doi:10.3390/jcm12031065_

Round 1

Reviewer 1 Report

This was a retrospective cohort study comparing outcomes of 179 term pregnancies with ventriculomegaly and 222 control pregnancies. Although this is an important topic, the study contains significant errors and inconsistencies, and the manuscript is not well written.

Major comments:

The authors declare: „Women who did not deliver at our hospital were excluded, and infants without follow-up after birth were excluded from the analysis. 22(10.95%) subjects in the VM group and 40(15.4%) subjects in the control group were lost to follow-up(P=0.176).“ From Table 1, it appears that obstetric outcomes are known for only 92 and 45 cases, respectively. If this was not found in the rest of the women, then they should have been excluded from the analysis. The cesarean section rates of 8.7% and 4.4% in term deliveries seem very optimistic…

Table 2 formatting is wrong.

Table 3 suggests there were no cases of mild VM, only moderate or severe... This seems highly unlikely.

Table 3 suggests there were 110 cases with abnormal chromosomal analysis from amniocenteses, but in table 1, there are only 11 cases…The number of cesarean sections and other parameters do not agree either.

Factors associated with abnormal neurological outcomes, according to table 3 are also non-isolated VM and bilateral VM but this is not mentioned in the abstract. Conversely, NICU admission and peak width were not more common in adverse neurological outcome VM, but this is wrongly stated in the abstract.

Extensive English language revision is also necessary.

Overall, I must conclude that the article is not of sufficient quality to be published in JCM.

Reviewer 2 Report

This is a retrospective study to investigate the short- and long-term outcomes of fetal ventriculomegaly. For this purpose, the authors identified 401 term pregnancies, 179 fetuses diagnosed of ventriculomegaly and 222 controls. The authors found that there are more chances of having a bad outcome in the ventriculomegaly group compared to the control one. Nevertheless, I have some questions and comments:

1)    Short-term outcomes are very poor. The authors should define the period they studied them.

2)    When the authors say long term outcomes, what is exactly the period they referred to? This should be specified in the text, along with a proper description of what test they used to study the infants and the kind of bad long outcomes.

3)    In the Design and Methods part, it is not clear to me the reason why the authors randomly selected the VM cases and not all the ventriculomegaly cases were included.

4)    It is not specified the different outcomes based on the level of ventriculomegaly.

5)    Table 2 should be corrected as the lines are not correlated with the variable.

6)    In the discussion part, it should be explained clearer why neonates with a ventriculomegaly a higher rate of admission in hospital, since Apgar score is similar in both groups as expected. 

7)    In line 192, references should be added.

Round 2

Reviewer 1 Report

Correcting the formatting of the tables has significantly improved the quality of the article. Nevertheless, I have a few reservations.

1) The authors state: „additional details on VM were recorded (i.e., whether bilateral or unilateral VM, first scanning ventricular width, gestational age of first scanning, peak

ventricular width and last scanning ventricular width).“, but some of these are not reported in the study. Also, it is not clear to me how the VMs were categorized into mild, moderate, or severe. Was it according to the first exam, according to the peak width, or according to the last measurement? Additionally, VM was not confirmed by the postpartum ultrasound in a significant number of cases. Can you comment on these?

2)  The authors state: „Furthermore, the pairwise comparison showed no statistical differences between mild VM and moderate VM (P>0.017), as well as between moderate and severe VM (P>0.017).“ I do not get why the p-value cutoff 0.017.

3) „While short-term outcome, especially NICU admission and peak width of the lateral ventricle, could predict their adverse long-term outcomes.“ Rewrite the sentence. If the aim of the authors was to find predictors of abnormal neurologic development, then logistic regression could have been considered.

4) Still I feel extensive English language editing is required. For example:

line 23: aiming to evaluate

line 24: „from“ instead of „of“

line 30: the lateral

etc.
